# Rho GTPases in Retinal Vascular Diseases

**DOI:** 10.3390/ijms22073684

**Published:** 2021-04-01

**Authors:** Akiyoshi Uemura, Yoko Fukushima

**Affiliations:** 1Department of Ophthalmology and Visual Science, Nagoya City University Graduate School of Medical Sciences, Nagoya 467-8601, Japan; 2Department of Cell Biology, National Cerebral and Cardiovascular Center Research Institute, Osaka 564-8565, Japan; 3Proteo-Science Center, Ehime University, Toon 791-0295, Japan; 4Uemura Eye Clinic, Nishinomiya 663-8101, Japan; 5Department of Ophthalmology, Osaka University Graduate School of Medicine, Osaka 563-0871, Japan; youko.fukushima@ophthal.med.osaka-u.ac.jp; 6Integrated Frontier Research for Medical Science Division, Institute for Open and Transdisciplinary Research Initiatives, Osaka University, Osaka 565-0871, Japan

**Keywords:** angiogenesis, cell–cell junction, cell migration, endothelial cell, retina, Rho GTPase, RhoJ, semaphorin 3E, vascular endothelial growth factor A, vascular permeability

## Abstract

The Rho family of small GTPases (Rho GTPases) act as molecular switches that transduce extrinsic stimuli into cytoskeletal rearrangements. In vascular endothelial cells (ECs), Cdc42, Rac1, and RhoA control cell migration and cell–cell junctions downstream of angiogenic and inflammatory cytokines, thereby regulating vascular formation and permeability. While these Rho GTPases are broadly expressed in various types of cells, RhoJ is enriched in angiogenic ECs. Semaphorin 3E (Sema3E) releases RhoJ from the intracellular domain of PlexinD1, by which RhoJ induces actin depolymerization through competition with Cdc42 for their common effector proteins. RhoJ further mediates the Sema3E-induced association of PlexinD1 with vascular endothelial growth factor receptor (VEGFR) 2 and the activation of p38. Upon stimulation with VEGF-A, RhoJ facilitates the formation of a holoreceptor complex comprising VEGFR2, PlexinD1, and neuropilin-1, leading to the prevention of VEGFR2 degradation and the maintenance of intracellular signal transduction. These pleiotropic roles of RhoJ are required for directional EC migration in retinal angiogenesis. This review highlights the latest insights regarding Rho GTPases in the field of vascular biology, as it will be informative to consider their potential as targets for the treatment of aberrant angiogenesis and hyperpermeability in retinal vascular diseases.

## 1. Introduction

Retinal vascular diseases are characterized by aberrant angiogenesis accompanying the dysregulated proliferation and migration of endothelial cells (ECs), which causes vitreous hemorrhages and, ultimately, tractional retinal detachment [1,2]. In these settings, the disruption of EC–EC junctions in preexisting and newly formed blood vessels causes plasma extravasation and retinal edema [3,4]. While various cytokines have been implicated in these pathologies, vascular endothelial growth factor (VEGF) A is a major driver of both angiogenesis and vascular hyperpermeability, which has been corroborated with the therapeutic potency of anti-VEGF agents in diabetic retinopathy, retinal vein occlusion, and retinopathy of prematurity [5,6,7]. In these diseases, retinal neurons, glial cells, and retinal pigment epithelium cells under hypoxia or inflammation upregulate expression levels of VEGF-A, which then binds to a tyrosine kinase VEGF receptor (VEGFR) 2 on ECs [5,6,7]. This ligand-receptor interaction induces the autophosphorylation of specific tyrosine residues in the VEGFR2 intracellular domain, activating downstream intracellular signaling cascades via extracellular signal-regulated kinase (Erk), Akt, and p38 [6,8,9]. In these events, the Rho family of small GTPases (Rho GTPases) play pivotal roles in the control of cytoskeletal rearrangements, cell–matrix adhesions, and cell–cell junctions in ECs, thereby regulating vascular formation and permeability [10,11].

Among the Rho GTPases, the best-studied are Cdc42, Rac1, and RhoA, which control EC behavior downstream of VEGF-A as well as various angiogenic and inflammatory cytokines [10,11]. Cdc42 and Rac1 promote polymerization of actin filaments underneath the cell membrane, enabling the extension of finger-like (filopodia) and sheet-like (lamellipodia) membrane protrusions, respectively [12,13]. RhoA induces stress fiber formation through the assembly of actin filaments with their motor protein non-muscle myosin II [12,13].

While Cdc42, Rac1, and RhoA are broadly expressed in various types of cells [14], a Cdc42 subfamily member RhoJ is enriched in angiogenic ECs [15]. Semaphorin 3E (Sema3E) releases RhoJ from the intracellular domain of a transmembrane receptor PlexinD1, by which RhoJ competes with Cdc42 for their common effector proteins and induces actin depolymerization [16,17]. In addition, RhoJ mediates Sema3E-induced VEGFR2-PlexinD1 association and p38 activation [17]. Upon stimulation with VEGF-A, RhoJ facilitates the formation of a holoreceptor complex comprising VEGFR2, PlexinD1, and neuropilin-1, thereby preventing the degradation of internalized VEGFR2 and prolonging downstream signal transduction events [17]. These pleiotropic functions of RhoJ promote directional EC migration in retinal angiogenesis both under physiological and pathological settings [17].

In this review, we summarize the accumulated knowledge of Rho GTPases in the field of vascular biology, with particular emphasis on RhoJ. This information will provide a clue to consider their potential as targets for the treatment of retinal vascular diseases.

## 2. Regulation of Rho GTPase Activity

Rho GTPases are small monomeric G proteins, which bind to and hydrolyze guanine nucleotides [12,13]. In vertebrates, 22 Rho GTPases are divided into eight subfamilies (Figure 1A) [12]. These Rho GTPases are tethered to plasma or endosomal membranes via their C-terminal CAAX motif, and cycle between a GTP-bound and GDP-bound state upon stimulation with extrinsic stimuli [12,13]. GTP-bound Rho GTPases recognize their effector proteins and generate a cellular response until they return to the GDP-bound state [12,13]. Thus, GTP- and GDP-bound Rho GTPases are assumed as active and inactive, respectively [12,13]. This cycle is regulated by three factors: guanine nucleotide exchange factors (GEFs), GTPase activating proteins (GAPs), and guanine dissociation inhibitors (GDIs) (Figure 1B) [12,13]. GEFs activate Rho GTPases by stimulating the dissociation of tightly bound GDP for replacement by more abundant GTP [13,18]. GAPs inactivate Rho GTPases by accelerating the intrinsically slow hydrolysis of GTP [13,18]. GDIs maintain Rho GTPases in an inactive state by altering their location from the cell membrane to the cytosol [13,19]. The RhoGDI family has only three members in mammals [13,19]. In contrast, approximately 80 RhoGEFs and over 60 RhoGAPs have been identified in humans [20], indicating that the activities of distinct Rho GTPases are regulated by multiple GEFs and GAPs in a context-dependent manner (Figure 2).

Cdc42 binds the Cdc42/Rac interactive binding (CRIB) domain of Wiskott–Aldrich syndrome protein (WASP) and neural WASP (N-WASP) [21,22]. Rac1 binds the CRIB domain of WASP-family verprolin homologous protein (WAVE) 2 [21,22]. WASP, N-WASP, and WAVE2 produce branched actin filaments by activating actin-related protein-2/3 (Arp2/3) [21,22]. Both Cdc42 and Rac1 bind the CRIB domain of p21-activated kinases (PAKs), which subsequently phosphorylate LIM kinase and cofilin, thereby blocking actin depolymerization [12,21]. Cdc42 and Rac1 also bind mammalian diaphanous (mDia)-related formins 2 and 3, which nucleate unbranched actin filaments, while RhoA binds mDia1 [12,21]. RhoA binds Rho-associated coiled-coil-containing protein kinases (ROCKs), which activate myosin light chain (MLC) kinase and inactivate myosin phosphatase, leading to the phosphorylation of MLC and the increase of actomyosin contractility [12,23]. ROCKs further activate LIM kinase and inactivate cofilin [12,23]. Thus, actin cytoskeletons are dynamically rearranged by spatio-temporal activation patterns of the Rho GTPases and their related effector proteins (Figure 2).

## 3. Regulation of Angiogenesis by Rho GTPases

### 3.1. EC Migration

At the sprouting edges of growing blood vessels, ECs migrate by sensing their microenvironments, extending the membrane protrusions, and reorganizing the cell–matrix adhesions [24]. During this sequence, the polarized subcellular localization of Rho GTPases contributes to the establishment of the front-rear axis in ECs (Figure 3) [24]. At the front of migrating ECs, VEGF-A activates Cdc42, thereby promoting linear actin polymerization and filopodia formation [24]. In contrast, repulsive cues such as Sema3E retract endothelial filopodia by inactivating Cdc42 and depolymerizing actin filaments [16]. At the proximal end of filopodia, VEGF-A activates Rac1, whereby the WAVE regulatory complex produces branched actin filaments in lamellipodia [24]. Notably, Arp2/3 and formins compete for actin polymerization; thus, Arp2/3-mediated lamellipodia formation blocks formin-mediated filopodia formation [25,26].

At the leading front of migrating ECs, the extending membrane protrusions adhere to the underlying extracellular matrix via integrins [24]. These focal adhesions are strengthened by the RhoA-ROCK-mediated phosphorylation of LIM kinase, the inhibition of cofilin-mediated actin depolymerization, and the enhancement of integrin clustering [27]. Rab13 transports RhoA and the GEF family member Syx to the leading front of migrating ECs, thereby targeting the RhoA-Syx complex to phosphorylated VEGFR2 [28]. In contrast, at the rear of migrating ECs, ROCK-mediated actomyosin contractility generates the pull-back force of membranes, leading to the release of focal adhesions [24]. Thus, RhoA regulates the dynamic remodeling of cell–matrix adhesions both at the front and the rear of migrating ECs.

### 3.2. EC–EC Junctions in Angiogenic Vessels

In growing blood vessels, individual ECs continuously move forward and backward, changing their respective positions [29,30]. In this process, ECs expand lamellipodia onto neighboring ECs, which then retract but retain connecting bridges of filopodia-like actin bundles where the vascular endothelial (VE)-cadherins condense (Figure 4) [31]. The nascent EC–EC connections are further strengthened by myosin-mediated stress fiber formation at the root of the connecting bridges [31]. Discontinuous VE-cadherins at the EC–EC junctions trigger the Rac1-Arp2/3-mediated formation of junction-associated intermittent lamellipodia, which generates VE-cadherin-based adherens junctions (AJs) [32,33]. Furthermore, Wnt5a-mediated Cdc42 activation stabilizes EC–EC junctions by promoting the binding of vinculin to the VE-cadherin/catenin complex, which is requisite for collective EC behavior in sprouting vessels [34]. On the other hand, VE-cadherin induces actomyosin contractility at cell–cell junctions via ROCK-dependent MLC phosphorylation, leading to the cessation of angiogenic sprouting [35]. Thus, Rho GTPases differentially contribute to the initial formation of EC–EC junctions in angiogenic blood vessels.

### 3.3. Retinal Vascular Development

Retinal vasculature develops radially from the optic disc to the periphery before birth in humans [2]. This process takes place after birth in mice, enabling the inducible EC-specific disruption of Rho GTPases during retinal angiogenesis [36]. Unexpectedly, endothelial RhoA deletion had no significant impacts on the retinal vascular development [37]. In contrast, the EC-specific deficiency of Rac1 and Cdc42 resulted in a significant reduction of retinal vascular growth [38,39,40,41]. At the cellular level, endothelial Cdc42 deficiency disrupted filopodia projections, the front-rear polarity, and VE-cadherin localization [41]. To date, however, information about the roles of Cdc42, Rac1, and RhoA in pathological retinal angiogenesis is less available. In the future, genetic manipulations of these Rho GTPases in ischemic retinopathy models will provide further insights into their roles in retinal vascular diseases.

## 4. Regulation of Vascular Permeability by Rho GTPases

Once the retinal blood vessels are formed, the association of ECs with pericytes facilitates the maturation of the EC–EC junctions and the suppression of transcytosis, which leads to the establishment of the blood–retina barrier (BRB) [42,43,44]. In retinal vascular diseases, however, the BRB integrity is broken down in pre-existing and newly formed blood vessels (Figure 5) [3,45]. In these settings, VEGF-A and tumor necrosis factor (TNF) α activate Rac1 via Vav2 and P-Rex1, respectively, leading to the PAK-mediated phosphorylation, internalization and degradation of VE-cadherin [46,47]. In addition, the RhoA-mediated contraction of radial actin bundles to which VE-cadherins are anchored exacerbates vascular hyperpermeability [48]. Moreover, RhoA disrupts N-cadherin-mediated adhesion between ECs and pericytes, which can be counteracted by the Rac1-mediated recruitment of N-cadherin to AJs [49].

The Ras-family small GTPase Rap1 stabilizes VE-cadherin-based AJs by activating Rac1 and Cdc42 via Vav2 and FGD5, respectively, and inactivating RhoA via Arhgap29 [50]. Angiopoietin-1 (Ang1), an agonist for a receptor tyrosine kinase Tie2 on ECs, also counteracts the pro-permeable effects of VEGF-A [51,52]. This effect of Ang1 can be partly explained by the RhoA-mediated activation of mDia1, which leads to the inhibition of Rac1-mediated VE-cadherin internalization [53]. In line with this notion, the RhoA GEF Syx is retained at the cell junctions by Ang1 but is translocated away from the junctions by VEGF-A [54]. However, another report argued that Ang1-mediated junctional stabilization requires the inactivation of RhoA by Rac1 and p190 RhoGAP, which inhibits actomyosin contractility at the AJs [55]. As Ang2 antagonizes the anti-permeable effects of Ang1, the dual inhibition of VEGF-A and Ang2 is expected to be a new strategy for the treatment of retinal vascular diseases [52].

## 5. Regulation of Directional EC Migration by RhoJ

### 5.1. Identification of RhoJ as an EC-Enriched Rho GTPase

In the early 2000s, RhoJ was initially identified as a Cdc42 subfamily member TC10-like or RhoT, which binds the CRIB domains of PAK, WASP, and N-WASP [56,57]. In addition to the control of actin cytoskeletons in fibroblasts and neuronal cells [56,57], RhoJ was implicated in adipocyte differentiation [58] and the early endocytic pathway of clathrin-coated vesicles [59]. In 2008, combined bioinformatic and expression analyses revealed high expression levels of RhoJ in ECs [60]. In 2011, three groups independently showed that endothelial RhoJ is upregulated by a transcription factor Ets-related gene [61], reduces focal adhesions and actomyosin contractility [62], and mediates Sema3E-induced actin depolymerization and filopodia retraction [16]. RhoJ was further shown to regulate the polarized endosomal trafficking of podocalyxin and α5β1 integrin to control lumen formation and fibronectin remodeling, respectively [63,64]. VEGF-A indirectly inactivates RhoJ via the EC-specific Cdc42 GEF Aehgef15 [65], although a distinct GAP for RhoJ has not yet been determined. RhoJ activation is further regulated by its palmitoylation and membrane localization mediated by glutamine synthetase [66].

### 5.2. Integration of VEGF-A and Sema3E Signals by RhoJ

In ECs, GTP-RhoJ binds the intracellular Rho-binding domain of PlexinD1 [17]. Sema3E stimulation releases RhoJ from PlexinD1, enabling actin depolymerization and focal adhesion disassembly through competition with Cdc42 for their common effector proteins (Figure 6) [16,17,67]. In addition to this cell-collapsing ability, PlexinD1-bound RhoJ promotes reverse EC migration away from Sema3E by facilitating PlexinD1-VEGFR2 association, VEGFR2 transphosphorylation at Y1214 (but not Y1175), and the activation of p38 (but not Erk and Akt) [17]. It is postulated that actin depolymerization and p38 activation may differentially occur at the rear and front, respectively, of migrating ECs under Sema3E stimulation. On the other hand, VEGF-A stimulation induces VEGFR2 autophosphorylation at both Y1175 and Y1214 [6,8,9]. In this signaling event, RhoJ facilitates the formation of a holoreceptor complex comprising VEGFR2, PlexinD1, and neuropilin-1, thereby preventing the degradation of internalized VEGFR2 and prolonging activation of Erk and Akt, but not p38 [17]. After a period of signal transduction, RhoJ is converted to the GDP-bound state and shifts from PlexinD1 to VEGFR2, by which VEGFR2 undergoes degradation [17]. Thus, RhoJ promotes forward EC migration under VEGF-A stimulation [17]. It should be noted that RhoJ enhanced directional EC migration under co-stimulation with inverse gradients of VEGF-A and Sema3E, indicating that RhoJ synergistically integrates these attractive and repulsive cues [17].

### 5.3. RhoJ in Retinal Angiogenesis

In the developing retinal vasculature, RhoJ is widely expressed in angiogenic ECs [16,17,68]. However, during vessel maturation, RhoJ expression is downregulated in ECs but maintained in a subset of neurons and vascular smooth muscle cells [17]. Both the global and EC-specific knockout of the *Rhoj* gene moderately reduced the radial extension and lateral branching of developing retinal vessels in postnatal mice [17,68]. In contrast, endothelial RhoJ deficiency remarkably suppressed aberrant angiogenesis in an ischemic retinopathy model in which neovascular tufts intensively expressed RhoJ [17]. EC-specific RhoJ overexpression also reduced developmental and pathological angiogenesis in the mouse retina [16,65]. The pleiotropic roles of RhoJ in VEGF-A and Sema3E signals may underlie the phenotypic similarity in its loss- and gain-of-function studies. In tumor models, endothelial RhoJ deficiency suppressed angiogenesis and exacerbated vascular permeability, which was partly ascribable to the enhancement of the RhoA-ROCK signal [69]. Consistently, a hybrid radiosensitizing nanoparticle functionalized by an anti-RhoJ antibody selectively inhibited tumor angiogenesis upon a low dosage of radiation [70]. The potential of RhoJ as an anti-tumor target is further supported by its expression in tumor cells per se, such as melanoma [71,72], gastric cancer [73], glioblastoma multiforme [74], and breast cancer [75].

## 6. Rho GTPases as Therapeutic Targets of Retinal Vascular Diseases

Considering the transient efficacy and potential adverse effects of anti-VEGF drugs for retinal vascular diseases [76,77,78], Rho GTPases and related signaling pathways may be alternative targets to treat aberrant angiogenesis and vascular hyperpermeability. However, the development of drugs that counteract Rho GTPases is challenging because of their intrinsic structural features, such as the limited binding pocket for an inhibitor to access, a high affinity for guanine nucleotide, and GTP availability in the micromolar range in cells [79,80]. Furthermore, it is difficult to target a specific cellular process because of the context-dependent signaling of Rho GTPases. Nonetheless, Rho GTPases are changing from undruggable to druggable targets by exploiting new approaches such as the inhibition of interaction between Rho GTPase and GEF, the blockade of nucleotide binding, the interference of localization to the membrane, the enhancement or mimicry of GAP activity, and the inhibition of downstream effector proteins [79,80].

To avoid systemic toxicity, the local manipulation of Rho GTPase signals is desirable. Currently, an ophthalmic solution of a small-molecule ROCK inhibitor, ripasudil hydrochloride hydrate, has been approved for the treatment of glaucoma in Japan and other Asian countries [81,82]. Experimentally, topical ripasudil significantly reduced the retinal thickness, neovascularization, and avascular areas in mouse models of ischemic retinopathy [83,84]. In addition, the clinical potency of ripasudil and another ROCK inhibitor fasudil has also been reported in diabetic macular edema [85,86]. In these settings, direct effects of ROCK inhibitors on neuroglial cells should be carefully monitored. In this respect, EC-restricted or -enriched proteins such as RhoJ and Arhgef15 may be ideal targets for specifically treating abnormal blood vessels in the eye.

## 7. Conclusions

Given the potency of Ras inhibitors for cancer treatment [87], pharmacological management targeting Rho GTPases may be practical for restoring vision in patients with retinal vascular diseases. In the past three decades, the functions of Cdc42, Rac1, and RhoA have been intensively investigated in non-vascular cells such as fibroblasts and neurons, which has been translated into the vascular research. RhoJ has been added to the repertoire of an essential signaling mediator of cell migration, cell–matrix adhesion, and endocytic trafficking in ECs. However, it remains unclear how intricate crosstalk between Rho GTPases and their GEFs, GAPs, and effector proteins regulate multi-cellular behavior during angiogenesis, such as collective EC migration and EC-pericyte association. By elucidating this important question, Rho-targeted therapies will open new avenues for rebuilding healthy blood vessels in retinal vascular diseases.

## Figures and Tables

**Figure 1 ijms-22-03684-f001:**
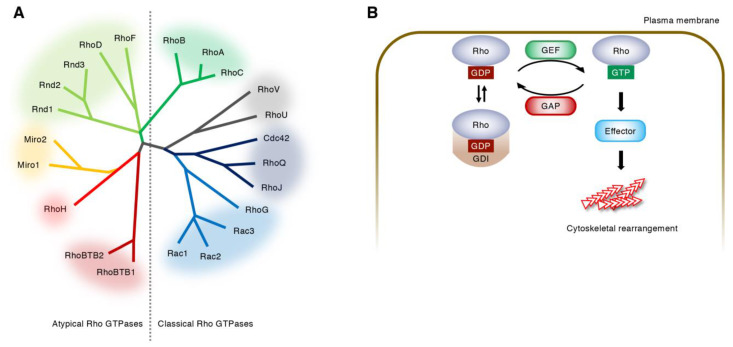
Regulation of Rho GTPase activity. (**A**) Rho GTPase family. Twenty-two Rho GTPases are divided into 8 subfamilies based on the amino-acid sequences. The atypical Rho GTPases are predominantly GTP-bound and are thought to be regulated by gene expression, protein stability, and phosphorylation. (**B**) Cycle of the classical Rho GTPases between a GDP-bound inactive state and a GTP-bound active state. Guanine nucleotide exchange factors (GEFs) catalyze the exchange of GDP for GTP, whereas GTPase activating proteins (GAPs) enhance the intrinsic GTPase activity of Rho GTPases. Guanine dissociation inhibitors (GDIs) sequester GDP-bound Rho GTPases in the cytoplasm.

**Figure 2 ijms-22-03684-f002:**
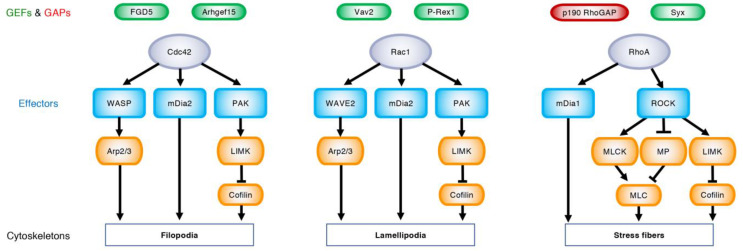
Upstream regulators and downstream targets of Rho GTPases. Under the regulation of specific GEFs (green) and GAPs (red), GTP-bound Cdc42, Rac1, and RhoA (purple) bind their effector proteins (blue) and rearrange actin cytoskeletons, leading to the formation of filopodia, lamellipodia, and stress fibers, respectively. Wiskott–Aldrich syndrome protein (WASP) and WASP-family verprolin homologous protein (WAVE) 2 produce branched actin filaments via actin-related protein-2/3 (Arp2/3). Mammalian diaphanous (mDia)-related formins nucleate linear actin filaments. p21-activated kinase (PAK) activates LIM kinase (LIMK), which phosphorylates and inhibits cofilin, thereby blocking actin depolymerization. Rho-associated coiled-coil-containing protein kinase (ROCK) phosphorylates myosin light chain (MLC) and induces actomyosin contraction through the activation of MLC kinase (MLCK) and the inactivation of myosin phosphatase (MP).

**Figure 3 ijms-22-03684-f003:**
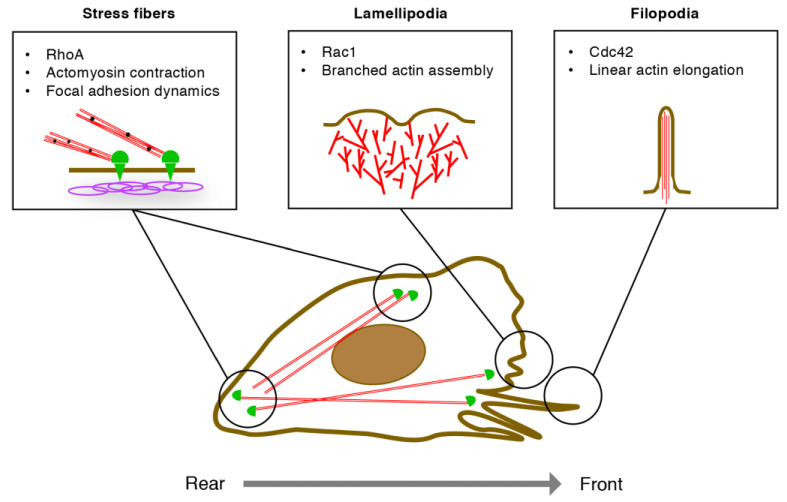
Regulation of endothelial cell (EC) migration by Rho GTPases. At the leading front, Cdc42 promotes linear actin elongation via mDia2, leading to filopodia formation. Rac1 assembles branched actin networks via WAVE2 complex and Arp2/3, leading to lamellipodia formation. RhoA promotes actomyosin contraction via ROCKs, whereby stress fibers control assembly and disassembly of focal adhesions at the front and the rear, respectively, of migrating ECs.

**Figure 4 ijms-22-03684-f004:**
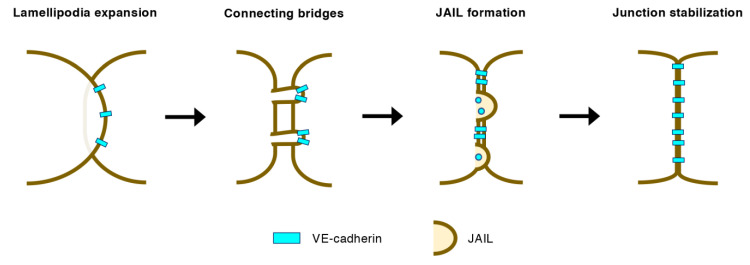
Initial formation of EC–EC junctions in angiogenic vessels. Lamellipodia expanding onto neighboring ECs retract but retain connecting bridges where vascular endothelial (VE)-cadherins condense. Discontinuous VE-cadherins trigger the Rac1-Arp2/3-mediated formation of junction-associated intermittent lamellipodia (JAIL), which subsequently generates VE-cadherin-based adherens junctions.

**Figure 5 ijms-22-03684-f005:**
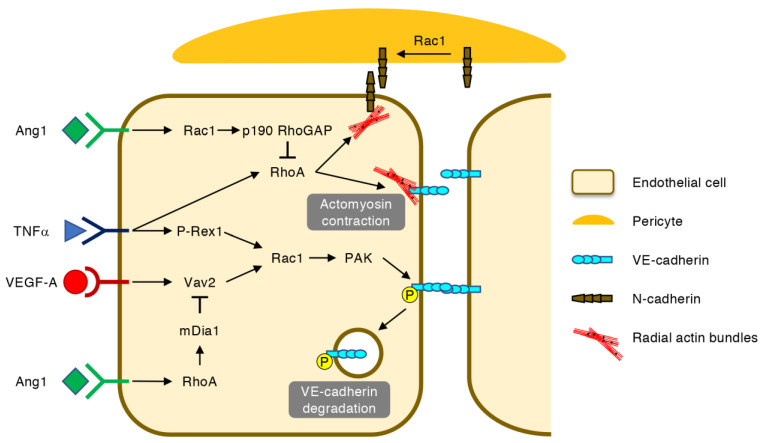
Regulation of vascular permeability by Rho GTPases. In ECs, vascular endothelial growth factor (VEGF) A and tumor necrosis factor (TNF) α activate Rac1 via Vav2 and P-Rex1, respectively, leading to the PAK-mediated phosphorylation, internalization and degradation of VE-cadherin. The RhoA-mediated contraction of radial actin bundles further destabilizes VE-cadherin-based EC–EC junctions and neural (N)-cadherin-based EC-pericyte junctions. Angiopoietin-1 (Ang1) prevents VEGF-A-induced VE-cadherin degradation via the RhoA-mDia1-mediated inhibition of Rac1 activation. In addition, Ang1 suppresses actomyosin contraction at the EC–EC junctions by inactivating RhoA via Rac1 and p190 RhoGAP. The Rac1-mediated recruitment of N-cadherin also protects EC-pericyte junctions.

**Figure 6 ijms-22-03684-f006:**
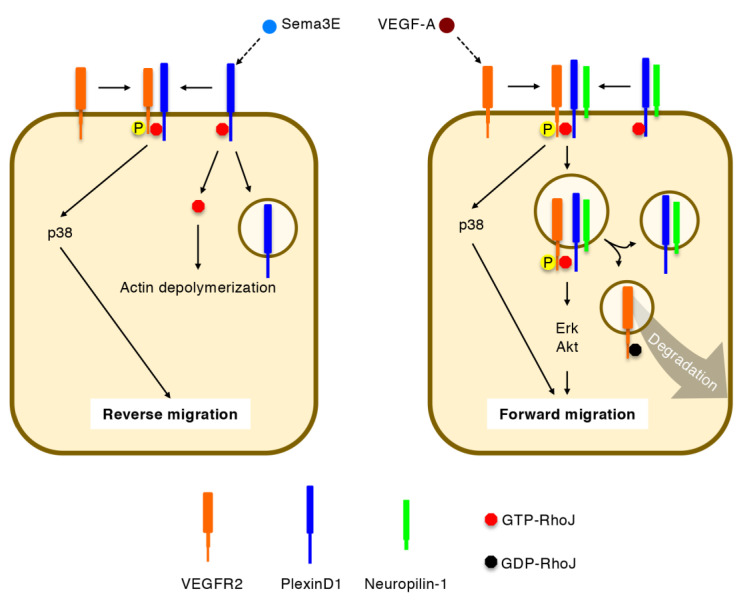
Regulation of directional EC migration by RhoJ. Semaphorin 3E (Sema3E) releases GTP-RhoJ from PlexinD1 to induce actin depolymerization. GTP-RhoJ further promotes Sema3E-induced VEGF receptor (VEGFR) 2-PlexinD1 association, VEGFR2 transphosphorylation, and p38 activation, thereby facilitating reverse cell migration. Upon VEGF-A stimulation, GTP-RhoJ mediates VEGFR2-PlexinD1-neuropilin-1 holoreceptor formation, prevents VEGFR2 degradation, and sustains signal transduction via extracellular signal-regulated kinase (Erk) and Akt, thereby facilitating forward cell migration. GDP-RhoJ promotes VEGFR2 degradation, thus terminating signaling via internalized VEGFR2.

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
