# Peer review of "Rho GTPases in Retinal Vascular Diseases"

_ijms, 2021, doi:10.3390/ijms22073684_

Round 1

Reviewer 1 Report

The present review provides a comprehensive description of the Rho GTPases involvement in angiogenic mechanisms and, particularly, their role in the signal transduction leading to the cytoskeletal remodeling and endothelial migration. The authors described the latest insight in the molecular mechanisms on which RhoGTPases differential activation exerts a polarizing effect for the directional EC migration. The review is well written and results potentially useful for the design of possible approach aiming to modulate angiogenic mechanisms involving RhoGTPases. However, some adjustment is needed to render the review adequately contextualized in ocular diseases.

  1. The title of the review refers to ocular neovascular diseases which comprises alteration of the vessels structure and function of several vascular plexuses of the eye. However, in the text, the authors focused the retinal angiogenesis involving retinal vascularization, which characterizes just a group of ocular neovascular disease. I would suggest adjusting the title and/or the text in order, at least, to consider other ocular neovascular disease not involving retinal vasculature.

  1. In paragraph 3.3, it is not clear whether the aim of the authors is to describe the role of Rho GTPases in the physiological formation of retinal vasculature or to also give an insight of their role in pathological conditions. Please clarify this point to improve the readability.

  1. The description of the role of Rho GTPases in ocular pathologies is limited to the RhoJ modulation. Please consider also providing a description of other RhoGTPases modulation in retinal and/or ocular diseases to draw a comprehensive state of the art of RhoGTPases and vascular disease of the eye.

  1. In the paragraph 6, the authors aimed to provide insights in the Rho GTPases as Therapeutic Targets Ocular Neovascular Diseases. The authors should consider the crucial role of the neurovascular unit which hinder the possibility to achieve a target-exclusive effect of the treatment on the vascularization. To date, this is one of the main limitations at the base of several long term adverse effects, as the case of anti-VEGF abolishing the neoangiogenic processes but also the neuroprotective activity of the growth factor. Since the role of RhoGTPases is crucial also for neurons, the authors should consider discussing the possible effect of the treatment on the neural component of the retina.

Reviewer 2 Report

The review paper  "Rho GTPases in Ocular Neovascular Disease" by Uemura and Fukushima, describes in detail the role of Rho GTPase in Ocular neovascular disease., angiogenic mechanisms and  their role in the signal transduction leading to the cytoskeletal remodeling and endothelial migration.

The review is well written, some minor changes in English language are needed. This study is thought to be able to give many researchers and medical scientists an understanding of the current research situation.
